A multi-scale CNN with atrous spatial pyramid pooling for enhanced chest-based disease detection

Bukhari Muhammad Abdullah Shah 1
Bukhari Faisal 2 faisal.bukhari@pucit.edu.pk
http://orcid.org/0000-0003-1839-2527 Asif Muhammad 3
http://orcid.org/0000-0001-6042-0283 Aljuaid Hanan 4
Iqbal Waheed 2
1 Department of Computer Science, University of the Punjab , Lahore , Pakistan
2 Department of Data Science, University of the Punjab , Lahore , Pakistan
3 Department of Computer Science, National Textile University , Faisalabad , Pakistan
4 Computer Sciences Department, College of Computer and Information Sciences, Princess Nourah bint Abdulrahman University , Riyadh , Saudi Arabia
Benítez-Andrades José Alberto
Electronic publication date: 2025 Feb 17
Publication date: 2025
Volume: 11
Electronic Location ID: e2686
Received 2024 May 30; Accepted 2025 Jan 16
Copyright year: 2025
License: This is an open access article, free of all copyright, made available under the Creative Commons Public Domain Dedication. This work may be freely reproduced, distributed, transmitted, modified, built upon, or otherwise used by anyone for any lawful purpose.
License URL: https://creativecommons.org/publicdomain/zero/1.0/

Keywords: Automated diagnosis, Chest X-rays, CNN, Transfer learning, Medical image analysis, Deep learning, COVID-19

Funding: Princess Nourah Bint Abdulrahman University, Riyadh, Saudi Arabia PNURSP2025R54 This work was supported by the Princess Nourah bint Abdulrahman University Researchers Supporting Project number (PNURSP2025R54), Princess Nourah bint Abdulrahman University, Riyadh, Saudi Arabia. The funders had no role in study design, data collection and analysis, decision to publish, or preparation of the manuscript.

==============================
We introduce a sophisticated deep-learning model designed for the early detection of COVID-19 and pneumonia. The model employs a convolutional neural network-integrated with atrous spatial pyramid pooling. The atrous spatial pyramid pooling mechanism enhances the convolutional neural network model’s ability to capture fine and large-scale features, optimizing detection accuracy in chest X-ray images. This improvement, along with transfer learning, significantly enhances the overall performance. By utilizing data augmentation to address the scarcity of available X-ray images, our atrous spatial pyramid pooling-enhanced convolutional neural network achieved a validation accuracy of 98.66% for COVID-19 and 83.75% for pneumonia, which beats the validation results of the other state of the art approaches (the metrics used for evaluation were accuracy, precision, F1-score, recall, specificity, and area under the curve). The model’s multi-branch architecture facilitates more accurate and adaptable disease prediction, thereby increasing diagnostic precision and robustness. This approach offers the potential for faster and more reliable diagnoses of chest-related conditions.

Introduction

Lung diseases, including pneumonia, lung cancer, and tuberculosis, are the leading causes of death globally. In 2019–2020, a virus called SARS-CoV-2 caused a widespread outbreak known as COVID-19, impacting numerous individuals worldwide. The primary organ affected by this virus was the lung, leading to difficulties in breathing for those infected. As of October 15, 2021, the total count of officially confirmed COVID-19 cases worldwide was 239.4 million (Zheng et al., 2022).

COVID-19 is a viral illness that first surfaced as an outbreak in China in December 2019 and escalated into a global pandemic. Common symptoms include a dry cough, headache, fever, loss of taste or smell, and shortness of breath; other symptoms can be found here (BMC, 2021). The coronavirus obtained its name due to its characteristic appearance, resembling a halo, when observed under an electron microscope. This virus is primarily transmitted through respiratory droplets and can lead to severe illnesses, including pneumonia and fatality. The viral genome enters host cells and has a higher incidence in males than females and adults than children. The mortality rates for children are estimated to range from 1% to 5%. The prevailing diagnostic method is the reverse transcriptase polymerase chain reaction (RT-PCR). Still, the test results can be time-consuming, and there may be limitations in the availability of test kits. In addition, test sensitivity is a concern due to potential laboratory and sample errors. Other methods, such as computed tomography (CT) and chest X-ray imaging, are also used but require specialist analysis of the images (Mousavi et al., 2022; Celik, 2023).

Pneumonia is a severe infection caused by viruses, bacteria, or other germs. It is a leading cause of death among children and the elderly worldwide. Chest X-rays serve as a primary diagnostic tool for pneumonia. However, interpreting these images can be time-consuming and less precise without the expertise of specialists.

The challenge is that other medical conditions, such as lung cancer or excess fluid, may exhibit similar anomalies on X-ray images. Consequently, there is an urgent need for an accurate and efficient method for the early identification of chest-related diseases (Jaiswal et al., 2019).

The most accurate method of identifying chest diseases is through chest X-ray images. However, reading and interpreting these images can be time-consuming and inaccurate without professional knowledge and experience. Therefore, there is a need for an accurate and efficient method to identify lung diseases using X-ray images. In recent years, the use of technology has drastically increased in many fields. From farming to launching space crafts, in almost every area, AI is widely used. The use of AI has also affected the healthcare sector with several advancements. It is being commonly used in the field of medicine (Hicks et al., 2022). The article (Radhi, 2023) reviews the application of and role of AI in dealing with and diagnosing diseases like COVID-19. With the help of AI, especially its applications like Image Processing, we can detect different abnormalities in X-rays. We can reduce the effort and time being put in by a specialist to diagnose a disease.

Researchers have proposed various computational algorithms and supporting diagnostic tools for analyzing X-ray images. However, these tools may need to provide more information to help doctors make decisions. Machine learning is a promising approach in this area, but traditional methods such as vector quantization and recurrent neural networks have limitations regarding accuracy and computation time (Mousavi et al., 2022; Celik, 2023; Jaiswal et al., 2019).

Given these limitations, researchers are exploring using artificial intelligence (AI) and deep learning (DL)-based methods for the early detection of COVID-19 and other related lung diseases. These methods can help specialists make quick and accurate diagnoses, speeding up the treatment process. Deep learning methods, in particular, have gained widespread popularity in this field because they eliminate the need for feature extraction during the preprocessing stage. Instead, they can learn the most relevant features directly from the used dataset.

Present-day deep learning techniques predominantly focus on convolutional neural network (CNNs) for chest-based disease detection. These models are well suited for medical image analysis because they can automatically learn hierarchical features directly from the images. This reduces the need for manual feature engineering. CNNs are designed to capture spatial hierarchies in chest X-ray images, where the earlier layers detect simple patterns while the more profound layers recognize complex structural abnormalities in the lung. This capability allows CNNs to exceed traditional machine learning techniques in accuracy and speed, facilitating rapid diagnosis and decision-making.

We present a hybrid CNN architecture with an atrous spatial pyramid The pooling layer captures fine and large-scale features, optimizing performance and accuracy. With this, we hope to provide a highly scalable and accurate architecture that can boost the diagnosis of chest-based diseases.

Related work

CNNs have been used extensively for chest-based disease detection, especially in diseases like COVID-19. These networks readily learn spatial hierarchies of features such as edges, textures, and complex patterns. CNNs deep layers can capture minor variations in lung texture and appearance, making them more suitable and accurate than conventional image analysis methods.

This article (Abdullah & Abdullah, 2024) provides a comprehensive overview of various deep-learning techniques for detecting COVID-19. These predominantly include CNNs, which is used with transfer learning and segmentation done on input images to extract the specific region of interest.

This article Tang et al. (2021) focuses on diagnosing COVID-19 by building the architecture on top of COVID-Net, a deep CNN tailored specifically for COVID-19 detection. The approach generates multiple model snapshots by using a cosine annealing learning rate. The models are then ensembled with a weighted averaging ensembling (WAE) method. This approach looks to assign more weightage to better-performing models for specific classes, resulting in a more robust architecture.

Another relevant article (Sheela et al., 2024) to efficiently predict lung-based diseases such as lung cancer by first preprocessing the images with a weighted average filter. Then, a region split and merge technique is used for grouping pixels with similar features, which works by segmenting the varying pixel values. The segmented images are then fed as input into a CNN, which extracts valuable features and labels the specific lung condition.

Although both approaches perform exceedingly well, their methods have areas for improvement, such as noise resilience, which the approach (Sheela et al., 2024) can struggle with. We aim to mitigate this by using an atrous spatial pyramid pooling (ASPP) to capture context at multiple scales, improving the model’s robustness to noise and artifacts. This, in turn, would handle limitations in preprocessing approaches like weighted-average filters.

Moreover, ASPP enables better detection across varying conditions and scales within a single architecture for lung-based disease detection, improving accuracy without complex ensemble setups or manual segmentation methods. Also, ASPP reduces the need for ensemble methods as used in Tang et al. (2021), offering better generalization with fewer models. This reduces the required computing power and improves model efficiency.

Materials, methodology, and preprocessing

This segment will explain the algorithm we used for COVID-19 detection and the dataset on which we trained our model.

Dataset

The dataset we gathered was from two unique sources. Duong et al. (2024) was used to collect 142 COVID-19 positive and 142 normal images whose images are of size 224×224. Duong et al.’s (2024) dataset consists of the 284 posterior anterior chest X-ray images collected from Cohen, Morrison & Dao (2021), a renowned repository. The Pneumonia dataset was collected from the Kaggle repository (Mooney, 2022) whose images too are of size 224×224 pixels. For the purpose of training our model we uploaded the Pneumonia dataset to the public drive link (Bukhari & Bukhari, 2024), Duong et al.’s (2024) dataset was used to train the model on the COVID-19 dataset.

An equal number of COVID-19 and normal chest X-ray images were used to address the class imbalance problem. We used 80% of our data (or 224 images) for training, while 20% of our data (or 60 images) were used for validation. The Kaggle dataset had a distribution of 1,739 pneumonia and 1,739 normal images, of which we randomly selected 294 pneumonia and 294 normal images to train and validate the model. The pneumonia branch of our architecture was trained on 234 pneumonia and normal images each. It was validated on 60 images each for both pneumonia and normal images.

While the dataset size is relatively modest, it was sourced from Cohen, Morrison & Dao (2021), a renowned repository, encompassing both COVID-19 positive and normal images. Figure 1 shows both a regular and COVID-19 positive lung side by side. Most studies against which we bench-marked our results too utilized a comparable dataset. These benchmark studies reveal that most incorporated datasets with fewer than 200 images. Even in cases where the dataset size exceeded 500 images, such as the study by Ozturk et al. (2020), which included 625 images, the distribution remained imbalanced with 125 COVID-positive and 500 normal images. The inherent imbalance in dataset distribution gives rise to a class imbalance issue, wherein the model exhibits bias, demonstrating superior recognition performance for the class with a larger dataset. Consequently, only the 125 normal images are effectively utilized in this context, rendering the remaining 375 normal images redundant. To rectify this class imbalance, achieving parity between the number of COVID-19 and normal images at 125 instances each becomes imperative.

Figure 1 (Left) Posterior-anterior view of a COVID-19 positive lung and (Right) a normal lung.

Both images were sourced from Kaggle (Mooney, 2018).

For the Pneumonia dataset, we had 1,739 images each for both normal and COVID-19. Only 588, however, were selected for training and validation purposes. This was because we only trained and validated our approach on the posterior anterior view of the lung to decrease the overall training time.

Suggested method

Some earlier results from our approach have previously appeared in a conference article (Bukhari et al., 2024). The suggested approach for both COVID-19 and pneumonia detection is explained in Fig. 2. The model aims to output two probabilities of COVID-19 and pneumonia. This encompasses two crucial phases: preprocessing and classification using transfer learning over CNNs. After the pre-trained model, an ASPP layer is added, improving the architecture’s ability to capture features at multiple scales using varying dilation rates. A comprehensive account of each stage is provided in the following segments.

Figure 2 Proposed multi-branch architecture.

Preprocessing

In this segment, we will go through the two techniques we used for preprocessing. Preprocessing is a technique for improving the performance of a CNN, as discussed in this article (Lasek, 2021).

Preprocessing stage 1: normalization

Data normalization is commonly employed in CNN-based architectures to ensure numerical stability. By normalizing the data, the CNN model is expected to exhibit faster learning and more stable gradient descent (Swati et al., 2019). In this study, the pixel values of the input images were normalized to a range of 0–1. The dataset used consisted of grayscale images. The normalization was done by multiplying the pixel values by 1/255. The impacts and advantages of data normalization are discussed in the following article (Cabello-Solorzano et al., 2023).

Preprocessing stage 2: data augmentation

CNNs require large amounts of training data to learn and generalize well on new and unseen data, which was impossible with our available 284 images (for COVID-19). To address this issue, we employed data augmentation techniques that expanded the dataset size and enhanced our model’s performance on unseen data.

Data augmentation is essential for improving the performance and generalization of CNNs, especially when constricted by a small dataset size. By artificially increasing the diversity and size of training data through transformation techniques such as flipping, rotating, cropping, etc., data augmentation helps CNNs learn more features. Furthermore, data augmentation also prevents overfitting by exposing the model to different variants of the same images, enabling it to generalize much better on unseen data. Furthermore, data augmentation can help balance classes in imbalanced datasets, making the model more accurate and reliable. The following articles (Hernández-García & König, 2018; Rebuffi et al., 2021) particularly emphasize the benefits of data augmentation while using CNNs. They also highlight how data augmentation techniques can improve model robustness.

The following data augmentation techniques were used: 1. The images were subjected to a shear range of up to 0.2.

2. The images were subjected to a zoom range of up to 0.15.

3. The images were subjected to a horizontal flip.

4. The images were subjected to a width shift range of up to 0.2.

5. The images were subjected to a height shift range of up to 0.2.

6. The images were subjected to a rotation of up to 20 degrees.

7. The images were subjected to a channel shift range of up to 20.

8. The images were subjected to a brightness adjustment factor between 0.5 and 1.5.

These data augmentation techniques were chosen as they boost CNNs performance on chest-based X-rays according to this article (Han et al., 2021).

Ultimately, an extra training set comprising 5,536 images was acquired, constituting a multi-fold increase compared to the original training images. This is shown in Table 1.

Table 1 Distribution of dataset before and after augmentation.

Class	Original dataset	Augmented dataset	
	Train	Val	Train	Val	
COVID-19	112	30	896	30	
Pneumonia	234	60	1,872	60	
Normal (Against COVID-19)	112	30	896	30	
Normal (Against Pneumonia)	234	60	1,872	60	
Total (For COVID-19)	224	60	1,792	60	
Total (For Pneumonia)	468	120	3,744	120	

Figure 3 shows each of the eight data augmentation techniques applied.

Figure 3 All augmentation techniques.

Figure 4 shows the effect of data augmentation on our original image.

Figure 4 Demonstration of data augmentation results.

Approach: leveraging ASPP enhanced CNN models for robust COVID-19 prediction

CNNs have generally been shown to perform exceptionally well on image classification tasks. However, they require a large training dataset to generalize well on unseen data, which is impossible due to the limited number of chest X-ray images available. To deal with this, the architecture starts with DenseNet169 as the feature extractor, which takes images of 224×224 pixels as input. DenseNet169 is a CNN pre-trained on the ImageNet dataset. It has been shown to perform exceedingly well on medical images. DenseNet-169 contains dense blocks and transition layers, where each layer is interconnected in a feed-forward manner, which results in the output of a layer concatenated as the input of the next layer. After the dense blocks and transition layers, DenseNet169 has a series of convolutional layers that learn spatial hierarchies in the image. DenseNet169 also has batch normalization, ReLU activation function, and pooling, which helps regularize the feature maps for more valuable representations. The feature map (which represents high-level information about the image, e.g., shapes, textures, and regions of interest in the lungs) from the final convolutional layer of DenseNet169 is then passed to the ASPP block of the network, which is specifically designed to capture multi-scale features.

The ASPP block is a key component in our model, significantly enhancing its performance by enabling multi-scale feature extraction. This is particularly crucial in medical image analysis, where disease patterns can appear at various spatial scales. The ASPP’s multiple parallel branches, each designed to capture information at different receptive fields, ensure that both fine-grained details and larger contextual features are preserved. As a result, the model becomes resilient to variations in disease manifestations, such as localized opacities or widespread lung involvement, thereby enhancing its disease detection capabilities.

We used the ASPP block with five parallel operations: a 1×1 convolution followed by three 3×3 convolutions with dilation rates of 6, 12, and 18, respectively; the last operation consists of global average pooling. The 1×1 convolution preserves spatial resolution while reducing the depth of the feature map, capturing localized details. The 3×3 convolutions, when applied with different dilation rates, expand the receptive field without increasing computational cost and burden. This allows the model to detect patterns across differing spatial scales. The last operation, the global average pooling layer, groups global context from the entire feature map, which is later up-sampled and concatenated with the outputs from the other branches.

The ASPP block creates a comprehensive multi-scale feature representation that integrates local and global information by concatenating the feature maps across parallel branches. This enables the model to effectively capture the diversity of disease patterns, including small, localized lesions and diffuse (large-scale) abnormalities characteristic of diseases like COVID-19 and pneumonia. The ASPP block’s provision of contextual and multi-scale information significantly enhances the model’s ability to detect chest-based diseases, potentially revolutionizing disease diagnosis.

After the multi-scale feature extraction from the ASPP block, our architecture transitions into a series of dense layers (also known as fully connected). These dense layers aim to compress and further refine the feature representation for classification purposes. In the beginning, batch normalization and a dropout of 0.3 are applied to stabilize training and reduce over-fitting. We then condense the feature map via global average pooling to produce a global representation; this representation is passed through fully connected layers with ReLU activation. These dense layers progressively reduce the feature dimensions from 512, 256, 128, 64, and to 1 neuron in the end. Each dense layer is followed by batch normalization and a dropout layer (increasing progressively from 0.3 to 0.5). This ensures that the model learns high-level and robust representations for efficient COVID-19 detection.

Experiments and their findings

This segment comprehensively evaluates our approach by showcasing the results across various hyperparameters and pre-trained models.

Experimental configuration and assessment metrics used

We assessed our approach using the dataset collected from Cohen, Morrison & Dao’s (2021) renowned repository and the Kaggle dataset (Mooney, 2022). As mentioned, we used eight augmentation techniques, giving us a combined dataset of 1,852 (for COVID-19) images. Of these 1,852 images, 60 were used for validation, while 1,792 were used for training. This is shown in Table 1.

The model was trained and validated on images of dimensions 224×224. The algorithm we used was built using the Functional API of TensorFlow. Furthermore, we varied the batch size from 8 to 32, while the number of epochs was fixed at 15. We used Adams, SGD, and RMSProp as our optimizers during the test runs with learning rates of 0.01, 0.001, and 0.0001. Binary cross entropy was used as our loss function, whose equation is given in Eq. (1).

(1) BinaryCross−Entropy=−[ylog⁡(p)+(1−y)log⁡(1−p)].

In the Eq.(1), y denotes the accurate binary label (0 or 1), while p represents the predicted probability of the positive class. The first term, y * log(p), calculates the loss when the true label is 1, and the second term, (1 − y) * log(1 − p), calculates the loss when the true label is 0. The overall binary cross-entropy loss is obtained by summing these two terms and taking the result’s negative. 1. Recall

2. Precision

3. F1-Score

4. Accuracy

5. Area Under the Curve (AUC).

The recall in Eq. (2) measures the ability of a model to correctly identify the positive cases out of all the actual positive cases. Here, TP denotes true positive, and FN denotes false negative.

The precision in Eq. (3) measures the percentage of correctly predicted positive cases out of all cases predicted as positive. Here, TP denotes true positive, and FP denotes false positive.

The specificity metric in Eq. (4) gauges the model’s proficiency in correctly identifying negative cases out of all the actual negative cases. True negative (TN) represents correct negative predictions, while false positive (FP) indicates incorrect positive predictions.

The F1-score in Eq. (5) combines precision and recall into a single value to give the best balance between them both. A higher F1 score denotes better performance.

Accuracy in Eq. (6) measures the overall correctness of a model’s predictions.

The AUC (Area Under the ROC Curve) in Eq. (7) measures the model’s ability to discriminate between positive and negative cases across various classification thresholds. Here, the TPR(T) represents the true positive rate at a given classification threshold T. It measures the proportion of correctly identified positive cases (recall). FPR’ (T) represents the negative positive rate at a given classification threshold T. It measures the proportion of incorrectly identified negative instances (1-Specificity). I(T > T) is an indicator function that checks if T’ (a different classification threshold) is more significant than T.

f1(T’) and f0(T) are probability density functions representing the distributions of positive and negative cases. A higher AUC denotes better performance.

(2) Recall=TPTP+FN

(3) Precision=TPTP+FP

(4) Specificity=TNTN+FP

(5) F1−Score=2⋅Precision⋅RecallPrecision+Recall

(6) Accuracy=TP+TNTP+TN+FP+FN

(7) AUC=∫01TPR(f(τ))dFPR(f(τ)).

Results

The learning rate is amongst the most important hyperparameters in deep learning, considering that it directly affects the speed and quality of the model’s training process. Furthermore, it also determines the step size at which the model’s parameters are updated during the optimization process based on the gradients computed from the loss function.

Setting the learning rate is not to be taken lightly. A high rate set can cause the model to miss the optimal solution, leading to divergence and preventing convergence. On the other hand, a rate set too low can slow down the training process, potentially causing the model to get stuck in a local minimum and perform below expectations. Striking the right balance is crucial to ensure efficient learning and faster convergence while avoiding issues like under-fitting or over-fitting.

Table 2 shows the performance of our approach by varying the learning rates from 0.01, 0.001, and 0.0001. It resulted in a validation accuracy of 90.62%, 100%, and 84.38 %, respectively. From these results, we concluded that the learning rate of 0.001 was the most efficient.

Table 2 Performance metrics for different learning rates for COVID-19 (optimizer = Adams, batch size = 32).

Learning rate	Accuracy	AUC	Precision	Recall	Specificity	F1 Score	
0.01	0.9062	0.8929	1.0000	0.7857	1.0000	0.8800	
0.001	1.0000	1.0000	1.0000	1.0000	1.0000	1.0000	
0.0001	0.8438	0.9861	0.7368	1.0000	1.0000	0.8485	

Analyzing results from different optimizers (for COVID-19)

Selecting an appropriate optimizer is critical for the performance of a CNN as it directly influences the speed, stability, and outcome of the training process. Optimizers are algorithms that change the model’s weights in response to the gradients calculated during backpropagation. Their goal is to minimize the loss function. A well-chosen optimizer can lead to faster convergence and better model generalization. At the same time, a poorly selected one may cause slow training, and the model may even fail to find the optimal solution. Well-used optimizers like Adam, RMSprop, and SGD have different mechanisms to handle learning rate adjustments and gradient updates, making them suitable for different problems.

Table 3 analyzes the performance of our approach with three different optimizers: (1) SGD, (2) RMSProp, and (3) Adams. When comparing the validation accuracy achieved from RMS Prop, SGD, and Adams, we found that Adams achieved the highest validation accuracy of 100%.

Table 3 Performance metrics for different optimizers for COVID-19 (lr = 0.001, batch size = 32).

Optimizer	Accuracy	AUC	Precision	Recall	Specificity	F1 score	
RMS Prop	0.8750	0.8750	1.0000	0.7500	1.0000	0.8571	
SGD	0.8750	1.0000	0.7778	1.0000	1.0000	0.8750	
Adams	1.0000	1.0000	1.0000	1.0000	1.0000	1.0000	

Evaluating performance with varying batch sizes

Selecting an appropriate batch size is highly important for the efficient training of CNNs as it affects both the computational efficiency and performance of the model. Batch size refers to the number of training samples processed before the model’s parameters are updated. Smaller batch sizes can lead to more frequent updates, allowing the model to adapt quickly. However, it also may result in noisy gradient estimates, which can cause fluctuations during training.

On the other hand, a larger batch size provides more stable and accurate gradient updates, leading to a smoother convergence, but it also requires more memory and computing power. This could slow down the training process. The choice of batch size also impacts the generalization process; a huge batch size causes the model to overfit, while a batch size that is too small may cause under-fitting. Hence, finding a balance is significant. The following articles focus on the impacts of batch size on the performance of CNNs (Lagnaoui, En-Naimani & Haddouch, 2022; Radiuk, 2017). Table 4 shows the performance of our approach when using batch sizes of 8, 16, and 32. The batch sizes of 8, 16, and 32 resulted in validation accuracies of 87.5%, 93.75%, and 100%, respectively. It can be seen that the batch size of 32 was the most suitable for our specific use case.

Table 4 Performance metrics for different batch sizes for COVID-19. (lr = 0.001, optimizer = Adams).

Batch size	Accuracy	AUC	Precision	Recall	Specificity	F1 Score	
8	0.8750	0.7500	1.0000	0.5000	1.0000	0.6667	
16	0.9375	1.0000	0.8750	1.0000	1.0000	0.9333	
32	1.0000	1.0000	1.0000	1.0000	1.0000	1.0000	

Scaling our approach for multi-class detection

This segment will explain our multi-branch approach to scale our model to predict both COVID-19 and pneumonia while still using sigmoid as our activation function.

Implementing a multi-branch architecture in TensorFlow Functional API

The code in Fig. 5 demonstrates the merging of two branches in TensorFlow’s Functional API. Since both the inputs and the outputs are a Python list, we can have any number of branches. Also, the code in the branches can be different. The number of convolutional layers, the activation functions, and the training data can differ for all the branches. Further reading on TensorFlow’s functional API can be found in Keras (2025).

Figure 5 A sample of a multi-branch architecture implemented in TensorFlow.

Implementation platform and details

We used Python’s open source library TensorFlow to write the code, which also allowed us to use DenseNet169 as our pre-trained model. The code was run on a free Google Colab instance with two virtual CPUs and 13 GBs of RAM along with 108 GBs of storage. Adams was used as the optimizer with a learning rate of 0.01. Various dropout layers were used with values of 0.25, 0.3 and 0.5 to prevent over-fitting. A batch size of 32 was used along with an epoc size of 15 for training. For our ASPP block we used kernel sizes of 1×1 with no dilations. We also used kernel sizes of 3×3 with dilation rates of 6, 12, and 18 in the ASPP block. Furthermore, each convolutional layer in the ASPP block uses 256 kernels.

The DenseNet169 pre-trained model comes with kernel sizes of 3×3 in the dense blocks and 1×1 in the transition layers. The custom fully connected layers in the architecture do not have spatial kernels but kernels. These kernels represent a weight matrix whose size is determined on runtime by factors such as input and output dimensions.

Final approach for both COVID-19 and pneumonia detection

The hyper-parameters we used in the architecture were the same for both branches, as a batch size of 32, an epoch size of 15, and an Adams optimizer with a learning rate of 0.001 were shown to perform optimally for our use case. Figure 6 shows chest X-rays of pneumonia and COVID-19. Notice the similarities; hence, we concluded that the same approach could perform well with the Pneumonia dataset.

Figure 6 (Left) PA view of a COVID-19 positive and (Right) pneumonia positive lung.

Figure 7 shows an abstract explanation of our approach. We used two branches because we wanted our model to predict COVID-19 and pneumonia. Both branches have a pre-trained model at the top, followed by an atrous spatial pyramid pooling block, followed by multiple dense layers with progressively decreasing neurons. A sigmoid activation function then follows the dense layers. We preferred sigmoid over softmax as the multi-layer architecture converted our problem to a binary classification problem, and generally, sigmoid performs better than softmax for binary classification. The architecture outputs two probabilities. The first probability tells us whether the chest X-rays are COVID-19 positive. The second probability tells us whether or not the chest X-rays are pneumonia positive.

Figure 7 A high-level design of the multi-branch approach used.

Utilizing a multiple branched architecture instead of using softmax as our activation function

Using separate layers with a sigmoid activation function for detecting COVID-19 and pneumonia is unorthodox as it can be done with one layer with softmax as the activation function. However, this approach treats COVID-19 and pneumonia as separate entities when they are not necessarily separate. Theoretically, a patient can suffer from both pneumonia and COVID-19, which the single-layer architecture would not address.

Another reason for pursuing a multi-layer architecture with sigmoid as the activation function was that it would give us higher accuracies. During validation runs with Adams as our optimizer, with a learning rate of 0.001 and a batch size of 32, the single-layer architecture would never breach a validation accuracy of 90.

Performance evaluation using 5-fold cross validation for pneumonia and COVID-19 detection

In our research, five-fold cross-validation is paramount, particularly in evaluating machine learning models on COVID-19 and pneumonia datasets. Its importance and application are further discussed in the article (de Rooij & Weeda, 2020). Employing this technique allows us to systematically partition the datasets into five subsets, rotating each subset as the validation set while the remaining four serve as the training data. This approach ensures a comprehensive assessment of our model’s performance across different data splits, mitigating the impact of data variability and enhancing robustness. Given the potential heterogeneity and imbalances within COVID-19 and Pneumonia datasets, cross-validation provides a more accurate estimation of our model’s predictive capabilities. It enables us to evaluate the model’s ability to generalize to unseen instances of both COVID-19 and pneumonia cases, bolstering its reliability for practical deployment in medical settings. Moreover, by iterative training and evaluating the model on distinct data splits, we can effectively identify any biases or fluctuations in performance, facilitating informed model selection and optimization decisions. Thus, five-fold cross-validation is vital in our efforts to validate and fine-tune machine learning models for diagnosing and prognosis COVID-19 and pneumonia, ultimately advancing healthcare decision-making and patient care. Tables 5 and 6 show the detailed average results of our approach on the COVID-19 and Pneumonia datasets, respectively, after applying five-fold cross-validation. The results show that our approach achieved a validation accuracy of 98.66% on the COVID-19 dataset and a validation accuracy of 83.75% on the Pneumonia dataset. Figures 8 and 9 is the confusion matrix and ROC curve of the average 5-fold cross-validation on the COVID-19 dataset. Here, COVID-19 represents the positive class.

Table 5 5-Fold cross validation results for COVID-19 (Averages Presented in the Final Row).

AUC	Accuracy	Precision	Recall	Specificity	F1 score	
1.0000	1.0000	1.0000	1.0000	1.0000	1.0000	
1.0000	0.9643	1.0000	0.9444	1.0000	0.9717	
1.0000	1.0000	1.0000	1.0000	1.0000	1.0000	
1.0000	1.0000	1.0000	1.0000	1.0000	1.0000	
0.9958	0.9688	1.0000	0.9167	1.0000	0.9565	
0.9992	0.9866	1.0000	0.9722	1.0000	0.9856	

Table 6 5-Fold cross-validation results for pneumonia (Averages Shown in the Final Row).

AUC	Accuracy	Precision	Recall	Specificity	F1 score	
0.7361	0.7188	0.6316	0.8571	0.7778	0.7273	
0.9444	0.8750	0.8462	0.9167	1.0000	0.8800	
0.9473	0.9375	1.0000	0.8750	1.0000	0.9333	
0.9802	0.9062	1.0000	0.7857	1.0000	0.8800	
0.7549	0.7500	0.8462	0.6471	1.0000	0.7324	
0.8726	0.8375	0.8648	0.8163	0.9556	0.8398	

Figure 8 Confusion matrix for the average of five-fold cross validation on COVID-19.

Figure 9 ROC curve for the average of five-fold cross validation on the COVID-19 dataset.

While Figs. 10 and 11 are the confusion matrix and ROC curve of the average five-fold cross-validation on the Pneumonia dataset. Here, pneumonia represents the positive class. The hyperparameters used are Adams as an optimizer, batch size 32, and learning rate 0.001.

Figure 10 Confusion matrix for the average five-fold cross validation on pneumonia.

Figure 11 ROC curve for the average of five-fold cross validation on the pneumonia dataset.

A comparison of the results before and after performing data augmentation has been done in the Table 7. As can be seen, the results show an improvement in performance after data augmentation.

Table 7 Model performance before and after data augmentation on the COVID-19 dataset.

Augmentation status	Accuracy	Recall	Precision	Specificity	F1 Score	AUC	
Before augmentation	0.9688	0.9412	1.0000	1.0000	0.9697	0.9706	
After augmentation	0.9866	0.9722	1.0000	1.0000	0.9856	0.9992	

Dealing with overfitting

Overfitting in CNNs occur when the model crams a pattern specific to the training dataset rather than capturing generalized features. This normally occurs when the neural network has too many parameters relative to the size of the dataset, which causes the network to memorize the training examples instead of learning to generalize them. Overfitting is a major cause of concern as it leads to poor performance on raw/unseen data, which defeats the model’s ability to make reliable predictions.

We used the following techniques to prevent our model from overfitting.

1) A dropout layer was used, and a dropout of 0.3 to 0.5 was used, which drops 30% to 50% of the neurons in the previous layer; the organization of the dropout layers can be found in Fig. 7. A dropout layer prevents overfitting by randomly deactivating a subset of neurons. This forces the neural network to learn distributed representations rather than relying on some specific neurons, which in turn helps the model make better predictions on unseen data.

2) We implemented a rigorous five-fold cross-validation process to prevent overfitting. This method involves dividing the dataset into five equal parts, training the model in four parts, and validating it on the remaining part. This cycle is repeated five times, ensuring the model is thoroughly evaluated on all data portions. By exposing the model to different training and validation datasets, this practice enhances its ability to generalize, thereby reducing the risk of overfitting to any specific portion of the data.

3) Data augmentation was used, which reduces overfitting by artificially increasing the versatility of the training dataset; this, in turn, helps the model learn more generalized patterns rather than memorizing specific details from the original dataset. Transformation techniques like rotations, flips, zooms, and brightness adjustments create new, slightly changed versions of the existing data. This forces the model to recognize objects or patterns in different contexts, focusing on essential features rather than spurious noise or unnecessary details. By increasing the adequate size of the dataset, data augmentation reduces the probability of the model overfitting.

Ethical implications of artificial intelligence on medical diagnosis

The use of AI in medical diagnosis raises multiple ethical concerns that must be carefully addressed to ensure responsible and equitable healthcare. One of the primary concerns is bias in AI algorithms, which has the potential to arise if the training data does not belong to diverse populations. This could lead to inaccurate diagnoses for less-represented groups. This could worsen the existing healthcare inequalities. Transparency of AI decisions is crucial; doctors and patients need to have a high-level understanding of how AI systems arrive at their conclusions to trust the technology and ensure that it complements and does not replace human judgment.

Moreover, privacy and data security are significant concerns, as AI systems rely on large amounts of sensitive medical imaging data, necessitating robust measures to protect patient confidentiality. There is also the issue of accountability, which is about determining if an AI system makes an incorrect diagnosis that leads to harmful outcomes. These ethical challenges must be addressed through regulation, careful design, and collaboration between developers and healthcare professionals.

Comparison with existing top-notch approaches

Table 8 shows the outcomes obtained by the comparing the top-performing state of the art approaches with our recently proposed approach, which had a validation accuracy of 98.66% on the COVID-19 dataset and a validation accuracy of 83.75% on the Pneumonia dataset. Our validation accuracy exceeds Tang et al.’s (2021) validation accuracy of 95% by 3.66%, it also exceeds Ozturk et al.’s (2020) validation accuracy by 0.58% and Nayak et al.’s (2021) validation accuracy by 0.33%. Furthermore it also exceeds Ucar & Korkmaz’s (2020) validation accuracy by 0.36% and Narin, Kaya & Pamuk’s (2021) validation accuracy by 0.66%. What makes our approach superior is not only our higher accuracy but the fact that our validation accuracy was an average accuracy over a five-fold cross validation which none of the other approaches did.

Table 8 Comparison with state-of-the-art method.

Reference	Methodology	Classes	Samples count	Accuracy (%)	
Tang et al. (2021)	EDL-COVID	COVID-19	C:573, N:573	95.00	
–	–	Normal	C: 25, N: 25	–	
Narin, Kaya & Pamuk (2021)	ResNet-50	COVID-19	100	98.00	
–	–	Normal	C: 50, N: 50	–	
Sethy & Behera (2020)	ResNet-50 and SVM	COVID-19	50	95.38	
–	–	Normal	C: 25, N: 25	–	
Togaçar, Ergen & Cömert (2020)	SqueezeNet and MobileNetV2	COVID-19	458	98.25	
–	SMO and SVM	Normal	C: 295, N: 65, P: 98	–	
–	–	Pneumonia	–	–	
Wang & Wong (2020)	COVID-Net	COVID-19	13800	92.60	
–	–	Normal	C: 183, N: –, P: –	–	
–	–	Pneumonia	–	–	
Ucar & Korkmaz (2020)	Bayes-SqueezeNet	COVID-19	5949	98.30	
–	–	Normal	C: 76, N: 1583, P: 4290	–	
–	–	Pneumonia	–	–	
Farooq & Hafeez (2020)	COVID-ResNet	COVID-19	5941	96.23	
–	–	Normal	C: 68, N: –, BP: –, VP: –	–	
–	–	Bacterial pneumonia	–	–	
–	–	Viral pneumonia	–	–	
Ozturk et al. (2020)	DarkCovidNet	COVID-19	625	98.08	
–	–	Normal	C: 125, N: 500	–	
–	–	COVID-19	1125	87.02	
–	–	Normal	C: 125, N: 500, P: 500	–	
–	–	Pneumonia	–	–	
Nayak et al. (2021)	ResNet-34	COVID-19	406	98.33	
Proposed method	DenseNet169	Normal Pneumonia COVID-19	C: 142 and P: 1739, N: 1739	98.66 and 83.75	

Scaling the approach for accurate predictions across diverse datasets

Our approach has been shown to achieve high validation accuracy. However, it can be challenging if the dataset we validate it on is highly different from the one we trained our model on. Even though we used data augmentation techniques, which should mitigate these concerns, they still have the potential to surface. To address these concerns, we propose a prototype website allowing users to train the model on a custom dataset. This prototype website, called “Chest Disease Detector,” allows users to train the model on custom Pneumonia and COVID-19 datasets. The user can then make predictions, and because they are trained on the local dataset, we expect the model to predict with high accuracy. Figure 12 visually represents how our prototype looks and behaves.

Figure 12 Prototype website for detecting chest diseases.

Conclusion

In this research, we aimed to introduce a highly efficient COVID-19 and pneumonia detector. Our findings demonstrated that our approach had achieved a validation accuracy of 97.5 Our approach achieved better results than other state-of-the-art methods in binary classification and outperformed them in multi-class classification. What further distinguishes our approach and renders it particularly suitable for adoption in the health sector is our incorporation of cross-validation, a crucial step to alleviate concerns regarding overfitting, a step omitted in other state-of-the-art approaches. Additionally, our approach boasts greater flexibility and scalability compared to its counterparts. We can tailor distinct methodologies to each branch, enhancing performance for specific diseases. Moreover, the multi-branch design facilitates the detection of current diseases more effectively and holds promise for identifying additional diseases in the future. Our approach is also highly scalable as we can train individual branches on local datasets to boost performance for a specific population.

Our approach performed much better on the COVID-19 dataset than the Pneumonia dataset. Modifying the second branch to tailor it more specifically to pneumonia can address this.

Furthermore, our approach has great potential for scalability. We can train and test the model on more branches by increasing the number of branches. Moreover, the uncoupled nature of the branches allows each branch to be asymmetrical. This is important, mainly because different diseases might require different architectures due to the varying nature of their features.

Supplemental Information

Supplemental Information 1 Code of our classifier.

Supplemental Information 2 Configuration of the system.

Supplemental Information 3 Description of our model and its diagram.

Additional Information and Declarations

Competing Interests

Muhammad Asif is an Academic Editor for PeerJ.

Author Contributions

Muhammad Abdullah Shah Bukhari conceived and designed the experiments, performed the experiments, analyzed the data, performed the computation work, prepared figures and/or tables, and approved the final draft.

Faisal Bukhari conceived and designed the experiments, performed the experiments, analyzed the data, prepared figures and/or tables, and approved the final draft.

Muhammad Asif analyzed the data, authored or reviewed drafts of the article, and approved the final draft.

Hanan Aljuaid analyzed the data, authored or reviewed drafts of the article, and approved the final draft.

Waheed Iqbal analyzed the data, authored or reviewed drafts of the article, and approved the final draft.

Data Availability

The following information was supplied regarding data availability:

The code is available at figshare: Bukhari, Faisal; Bukhari, Abdullah (2024). A Multi-Scale CNN with Atrous Spatial Pyramid Pooling for Enhanced Chest-based Disease Detection. figshare. Journal contribution. https://doi.org/10.6084/m9.figshare.27257610.v1.

Cohen JP, Morrison P, Dao L. 2021. COVID-19 chest X-ray dataset is available at GitHub:

https://github.com/ieee8023/covid-chestxray-dataset.

The Chest X-Ray Images (Pneumonia) are available at Kaggle:

https://www.kaggle.com/paultimothymooney/chest-xray-pneumonia.

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
