# Peer review of "A multi-scale CNN with atrous spatial pyramid pooling for enhanced chest-based disease detection"

_PeerJ Computer Science, doi:10.7717/peerj-cs.2686_

## Round 0.1 · original submission · Major Revisions

Dear authors,

Thank you for submitting your manuscript titled "Improving Automated Diagnosis of COVID-19 and Pneumonia from Chest X-ray Images through Convolutional Neural Networks and Transfer Learning" to PeerJ Computer Science. The manuscript has been thoroughly reviewed by several experts in the field, and their feedback has now been compiled.
After careful consideration, I regret to inform you that your manuscript requires major revisions before it can be considered for publication. While the topic is relevant and the approach has potential, the reviewers have identified several critical areas that need substantial improvement to enhance the quality, clarity, and novelty of your work.

Summary of Required Revisions:
1. Novelty and Contribution: Multiple reviewers have raised concerns about the lack of novelty in your approach, particularly in using pre-trained models and well-established methodologies. You are encouraged to more clearly articulate the novel contributions of your work and differentiate it from existing studies. Additionally, consider performing a more detailed comparative analysis with state-of-the-art methods to highlight the unique aspects of your approach.
2. Experimental Design and Methodology: The experimental design and methodology need to be more robustly described. This includes:
o Providing detailed explanations of the data augmentation techniques used and justifying their selection.
o Expanding on the technical details of your model, including architecture, hyperparameters, and evaluation metrics.
o Including additional analysis, such as confusion matrices, ROC curves, and statistical significance tests, to strengthen the validation of your results.
3. Literature Review and Context: Reviewers suggested expanding your literature review to include more recent and relevant works in the field. This will help position your research within the broader context of existing studies and underline the importance of your contributions.
4. Results and Discussion: There are suggestions to improve the clarity and depth of the results section. Reviewers noted potential overfitting issues and advised exploring regularization techniques like dropout or L1/L2 regularization. Moreover, discussing the generalizability of your model and its application across different datasets and demographics will strengthen your findings.
5. Clarity and Organization: The overall clarity and organization of the manuscript could be improved. This includes re-writing the abstract to focus more on the technical aspects, refining the introduction and conclusion sections, and ensuring that the paper is well-structured with a logical flow.
6. English Language and Presentation: Reviewers pointed out issues with grammar, tense, and sentence structure. It is strongly recommended to have your manuscript professionally proofread to ensure it meets the language standards required for publication.

Next Steps:
Please revise your manuscript in accordance with the feedback provided and submit a detailed response letter outlining the changes made, or providing justifications where changes were not made. Given the extent of the revisions needed, this process may take some time, but it will greatly improve the chances of your manuscript being accepted for publication.

We look forward to receiving your revised manuscript and the accompanying response to reviewers.

Thank you for your continued interest in PeerJ Computer Science.
Best regards,

Reviewer 1 ·

Basic reporting

Well-written manuscript

Experimental design

This research topic and methodology has been published by many research groups all over the world. There is no novelty in this study.

Figure 2 – It is impractical to have 2 branches where each branch only did a binary classification. You should do a multi-class classification with your datasets.

Validity of the findings

The authors have done the performance evaluation.

Additional comments

NIL

Reviewer 2 ·

Basic reporting

This article is using pre-trained models of deep learning on a public dataset which is available on the Kaggle. The authors are putting so much emphasis on the augmentation part of the dataset. I do not think there is any novelty involved in this article. There are many good works available in the state-of-the-art related to this COVID-19 outperforming the results of this article.

Experimental design

There is no contribution from authors when it comes to the experimental design. Only pre-trained models used.

Validity of the findings

Validation part of the models is weaker. I suggest authors to make used of more parameters.

Comparison with the state-of-the-art is not covered with the related works that outperform this work.

Check work: https://ieeexplore.ieee.org/document/9350186

Reviewer 3 ·

Basic reporting

Introduction
Literature Review: Enhance the introduction by incorporating a more thorough review of existing methods, including statistical methods and other machine learning techniques, to highlight the necessity and novelty of your approach.
Problem Statement: Clearly define the specific challenges or gaps your method addresses that previous methods have not.

Methodology
Data Augmentation: While data augmentation is used to address the dataset's limitations, the paper should specify which augmentation techniques were applied and justify their choice based on their impact on the model's learning capabilities.
Model Selection and Comparison: The selection of DenseNet169 as the superior model is based on its performance metrics, but the manuscript would benefit from a more detailed comparative analysis, including confusion matrices and ROC curves for each model tested.
Activation Function: The use of Sigmoid activation in a multi-class setting is unconventional as Softmax is typically preferred for such cases. A justification for this choice should be provided, or consider revisiting the activation function.

Results
Statistical Significance: Include statistical tests to assess the significance of the observed differences in model performances.
Overfitting: Discuss potential overfitting given the high accuracies and explore methods such as dropout or L1/L2 regularization to mitigate this.
External Validation: The robustness of the model can be further validated by testing it on external datasets not used in the training phase.

Discussion
Generalizability: Address the model's generalizability across different demographics and imaging equipment, which can vary widely in real-world settings.
Alternative Approaches: Consider discussing and comparing alternative deep learning architectures that might offer benefits over CNNs, such as Capsule Networks or Transformers.
Ethical Considerations: Introduce a discussion on the ethical implications of AI in medical diagnosis, particularly concerning data privacy and the potential for bias in AI models.

Conclusion
Future Work: Outline specific areas of future research, such as integration with other diagnostic tools or expansion to other types of diseases.
Collaborations: Suggest potential interdisciplinary collaborations that could enhance the research, such as with radiologists or data scientists specializing in health informatics.

Experimental design

Introduction
Literature Review: Enhance the introduction by incorporating a more thorough review of existing methods, including statistical methods and other machine learning techniques, to highlight the necessity and novelty of your approach.
Problem Statement: Clearly define the specific challenges or gaps your method addresses that previous methods have not.

Methodology
Data Augmentation: While data augmentation is used to address the dataset's limitations, the paper should specify which augmentation techniques were applied and justify their choice based on their impact on the model's learning capabilities.
Model Selection and Comparison: The selection of DenseNet169 as the superior model is based on its performance metrics, but the manuscript would benefit from a more detailed comparative analysis, including confusion matrices and ROC curves for each model tested.
Activation Function: The use of Sigmoid activation in a multi-class setting is unconventional as Softmax is typically preferred for such cases. A justification for this choice should be provided, or consider revisiting the activation function.

Results
Statistical Significance: Include statistical tests to assess the significance of the observed differences in model performances.
Overfitting: Discuss potential overfitting given the high accuracies and explore methods such as dropout or L1/L2 regularization to mitigate this.
External Validation: The robustness of the model can be further validated by testing it on external datasets not used in the training phase.

Discussion
Generalizability: Address the model's generalizability across different demographics and imaging equipment, which can vary widely in real-world settings.
Alternative Approaches: Consider discussing and comparing alternative deep learning architectures that might offer benefits over CNNs, such as Capsule Networks or Transformers.
Ethical Considerations: Introduce a discussion on the ethical implications of AI in medical diagnosis, particularly concerning data privacy and the potential for bias in AI models.

Conclusion
Future Work: Outline specific areas of future research, such as integration with other diagnostic tools or expansion to other types of diseases.
Collaborations: Suggest potential interdisciplinary collaborations that could enhance the research, such as with radiologists or data scientists specializing in health informatics.

Validity of the findings

Introduction
Literature Review: Enhance the introduction by incorporating a more thorough review of existing methods, including statistical methods and other machine learning techniques, to highlight the necessity and novelty of your approach.
Problem Statement: Clearly define the specific challenges or gaps your method addresses that previous methods have not.

Methodology
Data Augmentation: While data augmentation is used to address the dataset's limitations, the paper should specify which augmentation techniques were applied and justify their choice based on their impact on the model's learning capabilities.
Model Selection and Comparison: The selection of DenseNet169 as the superior model is based on its performance metrics, but the manuscript would benefit from a more detailed comparative analysis, including confusion matrices and ROC curves for each model tested.
Activation Function: The use of Sigmoid activation in a multi-class setting is unconventional as Softmax is typically preferred for such cases. A justification for this choice should be provided, or consider revisiting the activation function.

Results
Statistical Significance: Include statistical tests to assess the significance of the observed differences in model performances.
Overfitting: Discuss potential overfitting given the high accuracies and explore methods such as dropout or L1/L2 regularization to mitigate this.
External Validation: The robustness of the model can be further validated by testing it on external datasets not used in the training phase.

Discussion
Generalizability: Address the model's generalizability across different demographics and imaging equipment, which can vary widely in real-world settings.
Alternative Approaches: Consider discussing and comparing alternative deep learning architectures that might offer benefits over CNNs, such as Capsule Networks or Transformers.
Ethical Considerations: Introduce a discussion on the ethical implications of AI in medical diagnosis, particularly concerning data privacy and the potential for bias in AI models.

Conclusion
Future Work: Outline specific areas of future research, such as integration with other diagnostic tools or expansion to other types of diseases.
Collaborations: Suggest potential interdisciplinary collaborations that could enhance the research, such as with radiologists or data scientists specializing in health informatics.

Additional comments

Introduction
Literature Review: Enhance the introduction by incorporating a more thorough review of existing methods, including statistical methods and other machine learning techniques, to highlight the necessity and novelty of your approach.
Problem Statement: Clearly define the specific challenges or gaps your method addresses that previous methods have not.

Methodology
Data Augmentation: While data augmentation is used to address the dataset's limitations, the paper should specify which augmentation techniques were applied and justify their choice based on their impact on the model's learning capabilities.
Model Selection and Comparison: The selection of DenseNet169 as the superior model is based on its performance metrics, but the manuscript would benefit from a more detailed comparative analysis, including confusion matrices and ROC curves for each model tested.
Activation Function: The use of Sigmoid activation in a multi-class setting is unconventional as Softmax is typically preferred for such cases. A justification for this choice should be provided, or consider revisiting the activation function.

Results
Statistical Significance: Include statistical tests to assess the significance of the observed differences in model performances.
Overfitting: Discuss potential overfitting given the high accuracies and explore methods such as dropout or L1/L2 regularization to mitigate this.
External Validation: The robustness of the model can be further validated by testing it on external datasets not used in the training phase.

Discussion
Generalizability: Address the model's generalizability across different demographics and imaging equipment, which can vary widely in real-world settings.
Alternative Approaches: Consider discussing and comparing alternative deep learning architectures that might offer benefits over CNNs, such as Capsule Networks or Transformers.
Ethical Considerations: Introduce a discussion on the ethical implications of AI in medical diagnosis, particularly concerning data privacy and the potential for bias in AI models.

Conclusion
Future Work: Outline specific areas of future research, such as integration with other diagnostic tools or expansion to other types of diseases.
Collaborations: Suggest potential interdisciplinary collaborations that could enhance the research, such as with radiologists or data scientists specializing in health informatics.

Author could enrich his work by citing recent and related work as follow :
Abdullah et al. "Covid-19 Diagnosis using Deep Learning Approaches: A Systematic Review"
https://mesopotamian.press/journals/index.php/MJAIH/article/view/461
This paper provides a comprehensive review of various deep learning methodologies applied specifically to COVID-19 diagnosis. Citing this paper would enrich your manuscript by framing your research within the broader context of existing studies, highlighting where your method diverges or aligns with current trends. It would also provide a robust comparison against other deep learning models that have been validated in similar settings, thereby strengthening the argument for the efficacy and innovation of your approach.
Sheela et al. "Machine learning based Lung Disease Prediction Using Convolutional Neural Network Algorithm"
https://mesopotamian.press/journals/index.php/MJAIH/article/view/409
This paper explores the application of convolutional neural networks in diagnosing different lung diseases, not limited to COVID-19. Referencing this study could broaden the scope of your manuscript's discussion about the adaptability and scalability of CNN architectures for medical imaging beyond COVID-19. It would also allow you to discuss the generalizability of your model to other pulmonary conditions, providing a more comprehensive analysis of its potential applications and limitations.

Reviewer 4 ·

Basic reporting

It was a pleasure to review your research, this is an interesting and valuable topic. It has the potential to be published in the future, after minor revisions, including:
-The paper tackles the important problem of early detection of COVID-19 and other lung diseases, which is crucial for effective healthcare management.
- The use of CNNs for medical image classification is not entirely novel, and similar approaches have been proposed in the literature.
- Provide a detailed description of the datasets used for training and evaluation, including their size, diversity, and sources.
- Compare the proposed method with other state-of-the-art methods for medical image classification, such as traditional machine learning algorithms or other deep learning architectures.
-Please tidy and polish all of the tables and figures with care. A paper needs to have consistent figures and tables throughout. Please ensure that every figure has a good resolution of 300 dpi.
-References to previous studies on Artificial Intelligence's Role in Combatting the COVID-19, such as those found https://doi.org/10.58496/MJAIH/2023/002 , should be added to the work. This would bolster the claim and allow readers more resources to go deeper into.

-English needs to be improved by correcting tense and grammatical errors.
-Potential future directions for this research should be discussed by the author in order to increase the impact of the paper.

Experimental design

- Compare the proposed method with other state-of-the-art methods for medical image classification, such as traditional machine learning algorithms or other deep learning architectures

Validity of the findings

-Please tidy and polish all of the tables and figures with care. A paper needs to have consistent figures and tables throughout. Please ensure that every figure has a good resolution of 300 dpi.

Reviewer 5 ·

Basic reporting

The overall impression of the technical contribution of the current study is reasonable. However, the authors may consider making necessary amendments to the manuscript for the study to be more comprehensible. It is desired to have more technical explanation of the model.

Experimental design

More detailed explanation is desired. Authors may provide the architecture/block diagram of the proposed model for better comprehensibility of the proposed model concerning various aspects of the proposed model.

Validity of the findings

Explain the results for better comprehensibility and more comparative analysis is desired.

Additional comments

1. The abstract must be re-written, focusing on the technical aspects of the proposed model, the main experimental results, and the metrics used in the evaluation. Briefly discuss how the proposed model is superior.
2. Additionally, method names should not be capitalized. Moreover, it is not the best practice to employ abbreviations in the abstract, they should be used when the term is introduced for the first time. For example AI, CNNs
3. The contribution of the current study must be briefly discussed as bullet points in the introduction. And motivation must also be discussed in the manuscript.
4. The overall organization of the manuscript is not discussed anywhere in the manuscript. Please add the same in the introduction section of the manuscript.
5. The literature section is missing. Authors are recommended to incorporate the same for better comprehensibility of the study.
6. More explanation of the proposed model is desired on technical grounds. Also add more technical details about the data augmentation performed.
7. What is the size of the input image that is considered for processing and the size of the kernels?
8. The important details, like the input/tensor/kernel size, must be discussed, and whether authors have used Stride 1 or Stride, 2 must be presented. What type of activation function is being used in the current study.
9. What is the learning rate considered and discuss the other hyperparameters. For better idea refer studies like Enhancing Medical Image Classification via Federated Learning and Pre-trained Model  
10. Authors may provide the graphs of hyperparameters like training loss, testing loss, training accuracy, and testing accuracy. (if possible)
11. What are the cases assumed as TP, TN, FP, FN (confusion matrix) in the current study.
12. Please discuss more on the implementation platform and the dataset details as two sub-sections in the manuscript.
13. By considering the current form of the conclusion section, it is hard to understand by the Journal readers. It should be extended with new sentences about the necessity and contributions of the study by considering the authors' opinions about the experimental results derived from some other well-known objective evaluation values if it is possible.
14. English proofreading is strongly recommended for a better understanding of the study. Few sentences are written in passive voice and it is also observed that few sentences stopped abruptly.

---

## Round 0.2 · Major Revisions

Dear Authors,

Thank you for your submission and the effort invested in revising your manuscript. While we appreciate your work, we regret to inform you that the current version of your manuscript cannot be accepted for publication at this time. Below, we provide a summary of the key reasons for this decision, based on the reviewers' evaluations and our own assessment:

1. Novelty and Contribution: Reviewer 2 raises significant concerns about the novelty of the proposed methodology. The combination of Kaggle datasets, preprocessing, augmentation, transfer learning, and classification is not sufficiently innovative for publication in its current form. Additionally, the "Proposed Multi-Branch Architecture" lacks a clear and substantial contribution to the field.

2. Experimental Validation: The manuscript does not include a robust comparison with state-of-the-art methods, as highlighted by Reviewer 2. For instance, there is no quantitative demonstration of how the proposed approach outperforms existing techniques in terms of accuracy, efficiency, or other relevant metrics.

3. Clarity in Dataset Usage: The dataset description is ambiguous, giving the impression that the authors collected the data themselves. This lack of transparency undermines the credibility of the experimental setup.

4. Presentation and Organization: While some reviewers noted improvements, issues remain in the manuscript's structure. Reviewer 2 notes the misnumbering of sections and suggests consolidating the article's organization into a single paragraph. These issues contribute to a lack of professionalism in the presentation.

5. Technical Details: As pointed out by Reviewer 5, important implementation details are missing, such as the discussion of hyperparameters, kernel size, and platform specifics. These omissions make it difficult for readers to reproduce the work.

We recommend a thorough revision of the manuscript to address the above issues. A significant enhancement in the novelty and experimental validation is essential for reconsideration. Furthermore, clear and precise descriptions of datasets, methods, and implementation details will strengthen the manuscript.

While two reviewers supported acceptance, their comments lacked depth and did not address the critical issues raised by the other reviewers. Hence, we believe a major revisions is the most rigorous decision at this stage.

We encourage you to address these concerns and consider resubmitting after substantial improvements.

Sincerely,

Reviewer 2 ·

Basic reporting

Thank you for submitting the revised version of the article. Although it looks much improved, however, I feel the novelty of this article is still at the same level. Some of the major concerns of this article are as follows:
1. No need to add the distribution of the article in the introduction section. It is a scientific document. You can add organization of the article in one paragraph only rather than bulleted numbers.
2. Check the section 2 and 3. Section 2 is put mistakenly as Section 3 inside the Methods and Materials.
3. In dataset section, authors are using dataset from the Kaggle and repository. However, this section gives an impression as data set is gathered by the authors and all their images are collected as a contribution of this article.
4. I do not agree with the authors while mentioning "Proposed Multi-Branch Architecture". Input Kaggle Images+ Preprocessing+ Augmentation + Transfer Learning (Pre-trained)+ Activation function+ Classification (Output) is not anything new in this article.
5. Major focus of this article in terms of results on using different optimizers, and varying batch sizes. Next applying transfer learning and cross validation. Nowhere in the article, the authors are comparing the final results of this article with the state-of-the-art and showing the amount of percentage by which their scores are outperforming existing works.
6. I am seeing confusion matrix with only 57 and 42 labels in Fig. 10. May I know the reason for this small number?

Experimental design

I feel this article is still weak in terms of experimental design.

Validity of the findings

The article has not been properly validated with the state-of-the-art.

Reviewer 3 ·

Basic reporting

auhtor hav respons to all my comments

Experimental design

auhtor hav respons to all my comments

Validity of the findings

auhtor hav respons to all my comments

Additional comments

auhtor hav respons to all my comments

Reviewer 4 ·

Basic reporting

The manuscript has taken good shape now. All the best!!

Experimental design

The manuscript has taken good shape now. All the best!!

Validity of the findings

The manuscript has taken good shape now. All the best!!

Additional comments

The manuscript has taken good shape now. All the best!!

Reviewer 5 ·

Basic reporting

The overall impression in promising but still there are some minor corrections.

Please discuss more on the implementation platform and the dataset details as two sub-sections in the manuscript.
Please add the discussion on the hyperparameters being used.
Please add the details of kernel size and the number of kernels being used.

Experimental design

Please discuss more on the implementation platform and the dataset details as two sub-sections in the manuscript.
Please add the discussion on the hyperparameters being used.
Please add the details of kernel size and the number of kernels being used.

Validity of the findings

No comment

---

## Round 0.3 · accepted · Accept

The authors have addressed all of the comments.
Congrats!

Reviewer 5 ·

Basic reporting

The manuscript in well structured and explained.

Experimental design

The experimental design is well explained to reproduce the results.

Validity of the findings

The results seems to be sound.